# Early Complications and Results of Preserflo MicroShunt in the Management of Uncontrolled Open-Angle Glaucoma: A Case Series

**DOI:** 10.3390/ijerph19148679

**Published:** 2022-07-16

**Authors:** Emil Saeed, Renata Zalewska, Joanna Konopińska

**Affiliations:** Department of Ophthalmology, Medical University of Bialystok, 15-276 Bialystok, Poland; renata-zalewska@wp.pl (R.Z.); joannakonopinska@o2.pl (J.K.)

**Keywords:** Preserflo MicroShunt, complications, glaucoma, intraocular pressure, minimally invasive glaucoma surgery

## Abstract

We analyze the surgical outcomes and early complications with their management of the Preserflo MicroShunt (Santen Pharmaceutical Co., Ltd., Osaka, Japan) at six-month follow-up. The study is conducted between March 2021 and May 2022. Best-corrected visual acuity (BCVA) logMAR, intraocular pressure (IOP), and changes in glaucoma medications are assessed. Thirty eyes of 30 patients (22 women [73.3%] and 8 men [26.7%]) are included. They are augmented with mitomycin C (MMC) 0.5 mg/mL (8 subjects) or MMC 0.2 mg/mL (22 subjects) intraoperatively. BCVA is significantly higher one day after the treatment than before the treatment (MD with 95% CI = 0.05 (<0.01; 0.30); *p* = 0.045) when analyzing all patients. Such dependency is not observed when analyzing only patients treated with MMC 0.2 or 0.5 mg/mL (*p* > 0.050 for both analyses). No other statistically significant differences are detected in the level of BCVA before and after treatment. Among the patients, overall IOP is significantly lower at each time point after treatment than before surgery (*p* < 0.001 for all analyses). Among patients augmented with MMC 0.2 mg/mL, the IOP level is also significantly lower at each time point after treatment than before treatment (*p ≤* 0.001 for all analyses). The same differences are observed among patients with MMC = 0.5—the IOP level is significantly lower at each time point after treatment than before treatment (*p* < 0.050 for all analyses). Five subjects (16.7%) require anti-glaucoma medications three months after the procedure. Early complications (hypotony, choroidal effusion, keratitis, hyphema, and bleb fibrosis) are observed in 46.7% of cases. Our early results show that Preserflo MicroShunt is safe and effective for lowering IOP; however, it is not free from transient complications.

## 1. Introduction

Glaucoma is the second leading cause of irreversible blindness worldwide [1]. From 2010 to 2050, the number of people in the US with glaucoma is expected to increase more than two-fold, from 2.7 million to 6.3 million, according to the National Eye Institute. Quigley and Broman estimated the global number of people with glaucoma at almost 80 million, of which 74% have open-angle glaucoma [1].

Increased intraocular pressure (IOP) remains a major risk factor for glaucoma; thus, most current treatments aim to reduce IOP. If target pressure levels are not achieved and/or the disease is progressing despite the use of drug combinations, the next step on the therapeutic stepladder is to reduce the IOP surgically [2,3] according to the European Glaucoma Society Guidelines [4].

Minimally invasive glaucoma surgery (MIGS) has been developed as a less-invasive alternative to traditional incisional surgeries [2,3,5]. Generally, MIGS procedures are dedicated to mild to moderate glaucoma patients, and they are not intended to replace the more invasive conventional filtering surgeries, which are usually used for moderate to advanced glaucoma cases. Currently, there is no single common, widely accepted definition of MIGS. In a workshop of the American Glaucoma Society and the US Food and Drug Administration (FDA) held in 2014, the term “minimally invasive glaucoma surgery” was characterized by the implantation of a surgical device intended to lower IOP via an outflow mechanism, with either an ab-interno or ab-externo approach, associated with very little or no scleral dissection [2]. Among MIGS, there are many devices with a different mechanism of action as follows: Schlemm’s Canal implants that correct conventional outflow such as iStent^®^, Hydrus^®^; suprachoroidal space implants that strengthen outflow via the unconventional route as follows: CyPass^®^, iStent Supra^®^; subconjunctival implants that bypass physiological aqueous drainage pathways as follows: XEN^®^ Gel Stent, Preserlo^®^ MicroShunt [6,7].

The Preserflo MicroShunt (also known as InnFocus^®^ MicroShunt), an ab-externo subconjunctival device, might be an option for replacing trabeculectomy [8]. The Preserflo MicroShunt is a trans-scleral device that shunts aqueous humor from the anterior chamber to a filtering bleb under Tenon’s capsule and the conjunctiva [9]. It is an ab-externo drainage system made of a unique, biocompatible, and flexible material (SIBS). The Food and Drug Administration accepted the premarket approval application for Preserflo in July 2020. The Preserflo MicroShunt is indicated for patients with primary open-angle glaucoma (POAG) in whom the IOP remains uncontrolled despite the maximum-tolerated medical therapy or when glaucoma progression warrants surgery. Compared to traditional filtering surgeries, such as trabeculectomy, the Preserflo is less invasive, involves a shorter surgery time, and has a quicker recovery time. Another very important feature is its high safety profile. Intraoperative complications are uncommon with the Preserflo. Potential complications include hyphema or malposition of the shunt, with occlusion by the iris or placement of the tip of the device close to the endothelium, increasing the risk of endothelial cell damage [9].

Early complications are rarely reported but include transient hypotony and transient choroidal effusion (which are usually spontaneously resolved by week 3 and week 12, respectively); iris-tube contact, hyphema, and exposed Tenon’s capsule [10]. No bleb leaks, infections, or serious long-term complications have been reported [11,12]. However, appropriate detection and management of perioperative and early postoperative complications have an impact on the surgical outcome. Long-term results may be related to perioperative and postoperative problems.

Here, we discuss the first experiences and assess the early results and complications associated with the use of the Preserflo MicroShunt (Santen Pharmaceutical Co., Ltd., Osaka, Japan) in the 6-month follow-up.

## 2. Materials and Methods

The study was conducted between March 2021 and May 2022 in the Department of Ophthalmology, Medical University of Bialystok, Poland. The research protocol was approved by the Bioethics Committee of the Medical University of Bialystok (APK.002.581.2021). The study was performed in accordance with the ethical standards of the Guidelines of the Declaration of Helsinki (2008, modified 2013). All participants gave informed written consent.

The inclusion criteria were open-angle glaucoma (primary or pseudoexfoliation) when target IOP was not achieved with the maximum tolerated IOP-lowering medications and detected glaucoma progression. Exclusion criteria were as follows: narrow-angle or angle-closure glaucoma, active inflammatory eye disease or lack of consent for surgery, lack of patient’s contest for the surgery, and participation in the study. Thirty eyes of the following 30 subjects were included in the study: 25 patients with insufficiently controlled POAG and 5 patients with pseudoexfoliation glaucoma (PXG). Twenty-six patients received the Preserflo MicroShunt alone and four received a combined procedure of the Preserflo MicroShunt with phacoemulsification. All patients received 0.2 or 0.5 mg/mL mitomycin C (MMC). Thirty Caucasian patients (22 women and 8 men) with a mean age 68.53 ± 10.24 years old were enrolled in the study.

### 2.1. Preoperative and Postoperative Examination

Results were collected as follow-up visits were performed as follows: at 1 day, 2 weeks (±1 day), 1 month (±3 days), 3 months (±5 days), and 6 months (±7 days) postoperatively and in some cases (described below) more often. Preoperative examination (1 week ± 3 days) included obtaining patient data (age, sex, anti-glaucoma medications, and surgery procedures). Basic procedures included IOP measurement using a Goldman applanation tonometer, best-corrected visual acuity (BCVA) using the Snellen chart and then converted to logMAR units, anterior segment assessment with funduscopic examination using a slit lamp, Goldmann three-mirror lens for gonioscopy assessed with Schaffer classification, and optical coherence tomography (OCT) of the optic nerve disc using the Spectralis spectral domain-OCT (Heidelberg Engineering, Heidelberg, Germany).

Postoperatively, BCVA, IOP, and number of IOP-lowering medications were assessed, and the anterior segment and fundus were examined at each control visit. Postoperative complications were identified as hypotony (defined as an IOP < 6 mmHg, as suggested by the World Glaucoma Association Guidelines [4]), choroidal effusion, hyphema, keratitis, and bleb-related complications, such as fibrosis. In cases with hypotony, B-scan ultrasonography was performed using Quantel Medical B Scan Compact Touch.

All antiglaucoma treatment was withdrawn after the surgery. When the target IOP was not achieved postsurgery it was readministrated according to the EGS guidelines [4]. After surgery, all patients were prescribed dexamethasone 4 times daily for four weeks and then tapered, and moxifloxacine 3 times daily for one week.

### 2.2. Surgical Technique

All surgical procedures were performed under local anesthesia by the same surgeon. Surgical success was defined as IOP ≥ 6 and ≤18, and at least 20% IOP reduction by 6 months after surgery.

The surgical technique was performed as previously described [8]. In brief, following retrobulbar anesthesia with 2% lidocaine, a fornix-based conjunctival flap (5 mm) was created in the superior quadrant and a deep sub-Tenon’s pocket was formed. The subconjunctival and sub-Tenon’s spaces were treated with three sponges soaked with MMC, close to the limbus for 2 min. The area was then carefully irrigated with a balanced salt solution (BSS). After scleral cautery and drying of the area, the location of the scleral tunnel was marked 3 mm from the limbus. A pocket was made at the marked point using a 1-mm slit knife, and a 25-G needle track was formed from the sclera into the anterior chamber, immediately anterior to the plane of the iris, avoiding distortion of the tissues. The Preserflo MicroShunt (Santen Pharmaceutical Co., Ltd., Osaka, Japan, 8.5-mm length, with a 350-µm outer diameter and a 70-µm lumen) was then inserted ab-externo into the anterior chamber. The shunt is made of poly (styrene-block-isobutylene-block-styrene)—SIBS (Figure 1).

Droplet outflow was observed at the distal end of the microshunt. If flow was not visualized, BSS was injected via paracentesis, or the shunt was flushed using a 23-G cannula. Tenon’s capsule and conjunctiva were repositioned over the device and were sutured using 8-0 Vicryl sutures (Figure 2 and Figure 3). Phacoemulsification was performed before microshunt implantation in patients who required cataract surgery.

### 2.3. Statistical Evaluation

The analyzes were conducted in the R statistical package, version 4.1.3 (R Foundation for Statistical Computing, Vienna, Austria), using the Rcmdr, Misc, and Rcompanion libraries. The assumed confidence interval was α = 0.05. Number and percent of characteristics were presented for qualitative variables, mean, and standard deviation or median with quartiles 1 and 3 were presented for quantitative variables depending on distribution. Normality of distribution was analyzed with Shapiro-Wilk’s test. Two dependent measurements were compared using Wilcoxon’s test (if at least one distribution was not normal) or paired t-Student’s test (if both distributions were normal). Mean/median differences with 95% CI were assessed for each pair of compared variables.

## 3. Results

Thirty eyes of 30 subjects were included in the study. The demographic data are summarized in Table 1.

### 3.1. Best-Corrected Visual Acuity

BCVA was significantly higher one day after the treatment than before surgery (MD with 95% CI = 0.05 (<0.01; 0.30); *p* = 0.045) when analyzing all patients. Such dependency was not observed when analyzing only patients treated with MMC 0.2 or 0.5 mg/mL (*p* > 0.050 for both analyses). No other statistically significant differences were detected in the level of visual acuity before and after treatment, Table 2.

### 3.2. Intraocular Pressure

Among the patients overall, IOP was significantly lower at each time point after treatment than before surgery (*p* < 0.001 for all analyses). Among patients treated with MMC 0.2 mg/mL, the IOP level was also significantly lower at each time point after treatment than before treatment (*p ≤* 0.001 for all analyses). The same differences were observed among patients with MMC = 0.5—the IOP level was significantly lower at each time point after treatment than before treatment (*p* < 0.050 for all analyses), Table 3.

Surgical success was defined as IOP ≥ 6 and ≤ 18, and at least 20% IOP reduction after 1 month postoperatively was achieved in 17 of 20 cases (85%).

A range of variables are presented in Table 4.

### 3.3. Early Postoperative Complications

#### 3.3.1. Hypotony and Choroidal Effusion

The first hypotonic patient, a 61-year-old phakic female with pseudoexfoliation glaucoma with an IOP of 22 pre-ops., taking latanoprost and brinzolamide drops before surgery, presented with an IOP of 4 mmHg 2 weeks postoperatively. The shallow anterior chamber was noticed but without choroidal effusion. She has been treated with MMC 0.5 mg/mL intraoperatively.

The second hypotonic case was a phakic 72-year-old female with pseudoexfoliation glaucoma and an IOP of 23 who underwent combined surgery. She was treated with MMC 0.2 mg/mL intraoperatively and preoperatively using the following combined drops: timolol–brimonidine. She had an IOP of 5 mmHg on the first day postoperatively, with a shallow anterior chamber without choroidal effusion.

The third hypotonic patient was an 86-year-old pseudophakic male with pseudoexfoliation glaucoma and an IOP of 26, treated with MMC 0.2 mg/mL. He was on four anti-glaucoma medications prior to the surgery (latanoprost, brimonidine, timolol, and dorzolamide). He presented an IOP of 3 mmHg on the first day and 2 weeks after surgery without choroidal effusion. One month after the procedure, IOP increased to 8 mmHg.

The fourth patient was an 84-year-old woman with POAG. She was pseudophakic with a history of an Ex-Press Mini Glaucoma Shunt and trabeculoplasty and an IOP of 27 pre-ops. The patient was on two anti-glaucoma medications preoperatively (latanoprost and acetazolamide tablets, 500 mg/day). She presented an IOP of 5 mmHg on the first postoperative day with a shallow anterior chamber, choroidal effusion, and decreased visual acuity. The dose of MMC, in this case, was 0.2 mg/mL. OCT showed hypotonic maculopathy.

The fifth hypotonic case was a pseudophakic 88-year-old male with moderate primary open-angle glaucoma treated with MMC 0.2 mg/mL. Prior to the surgery, the patient was on four drops and their IOP was 20 mm Hg. He had an IOP of 3 mmHg on the first day postoperatively, with a shallow anterior chamber without choroidal effusion. One month postop, the IOP was 5 (still without choroidal effusion) and stabilized at 3 months after the surgery—16 mm Hg. Six months after the procedure, IOP increased to 23 and one drop was prescribed.

Intraocular pressure increased within 7 days in the first three cases but remained low until 4 weeks in the fourth case. What is striking is that the second and third patients experienced a significant IOP increase after 2 and 4 weeks, respectively (to 20 mmHg and 24 mmHg). In these cases, bleb needling, performed in a standardized way in the operating room, was performed with a subconjunctival injection of 40 µg MMC, and IOP decreased to 14 and 18, respectively.

#### 3.3.2. Hyphema

Hyphema grade III (according to the system of Edwards and Layden) significantly decreased visual acuity and an IOP of 7 mm Hg was noticed in a 66-year-old phakic man with advanced POAG and bipolar disease. This was probably due to taking clopidogrel after a heart attack. In this case, MMC 0.2 mg/mL was used. We did not observe any vitreous hemorrhage upon ultrasound. He was advised to use a half-sitting position and topical cycloplegic agents and steroids were administered. With frequent hospital visits, blood was absorbed within 2 weeks with a BCVA of 0.4 (logMAR) and an IOP of 10. Three months after the glaucoma procedure, he underwent phacoemulsification and IOP increased to 27 mmHg on the first day, 22 mmHg two weeks postoperatively, and decreased to 14 mmHg one month after cataract surgery.

#### 3.3.3. Keratitis

Two phakic POAG patients, a 72-year-old male and a 58-year-old female, developed keratitis lesions (1 × 1 mm) close to the bleb one day post-op. They both received the MicroShunt alone with 0.5 mg/mL MMC intraoperatively. Topical steroids were reduced, and a bandage contact lens was administered. Keratitis healed within 7 days and visual acuity improved. These complications, probably related to the higher dose of MMC, led us to reduce the MMC dose to 0.2 mg/mL.

#### 3.3.4. Bleb Fibrosis

A patient with an IOP increase one month after surgery was encountered. A 57-year-old phakic woman with POAG had an IOP of 8 mm Hg on the first day and reached 27 mmHg one-month post-op compared to 22 mmHg preoperatively. The dose of MMC was 0.2 mg/mL. She qualified for the needling procedure, but the result was poor. The IOP decreased to 23 mmHg and the patient was then qualified for bleb revision. The revision was successful and IOP decreased to 8 mm Hg on the first day and seven days after surgery. Three months after the surgery, IOP increased to 23 mmHg and one medication was then prescribed.

Another patient, a 72-year-old woman with an IOP of 23 pre-op, underwent the combined procedure with MMC 0.2 mg/mL. She developed hypotony on the first day and required bleb needling two weeks after the surgery as IOP increased to 23 mmHg. One month after the procedure, IOP decreased to 14 mm Hg and stayed stable. In this case, we also observed an intense flare in the anterior chamber. Early postoperative complications and interventions are summarized in Table 5.

Five subjects (16.7%) required anti-glaucoma medications three months after the procedure.

## 4. Discussion

Microshunt implantation is increasingly gaining popularity as it is an effective procedure for reducing IOP. In this study, we focused on early complications and management after the Preserflo MicroShunt. Subjects received MMC close to the limbus at a concentration of 0.2 or 0.5 mg/mL, as better results were expected after a higher dose [12]. However, we found no significant difference in BCVA before and after treatment among patients who received MMC at 0.5 mg/mL or 0.2 mg/mL. All patients’ IOP levels were significantly lower at each time point after treatment.

Recently, a landmark study directly compared the MicroShunt with trabeculectomy in a randomized controlled trial [13]. The most common complications observed by the authors were increased IOP, hypotony, subconjunctival bleeding, and bleb leak; however, these complications occurred more often in the trabeculectomy than in the MicroShunt group. The rate of sight-threatening complications was low in both groups (IOP spike > 10 mmHg from baseline, malignant glaucoma, and choroidal effusion requiring surgical treatment occurred in 1.0% in the MicroShunt group and 0.8% in the trabeculectomy group).

The concentration and duration of MMC exposure affect the surgical success of glaucoma filtration surgery [14]. The concentration of MMC used during trabeculectomy ranges from 0.1 mg/mL to 0.5 mg/mL [15]. Riss et al. showed that the efficiency of the MicroShunt depended on the placement and concentration of MMC. Patients treated with 0.4 mg/mL MMC placed close to the limbus presented a 55% reduction in IOP, while eyes with the same concentration of MMC placed deep in the pocket presented a 32% reduction in IOP [16]. MMC is a powerful tool for wound modulation if it is applied to enhance the results of filtering surgery for glaucoma. However, its alkylating and antifibrotic properties exert toxic effects on corneal endothelial cells, keratocytes, and limbal stem cells [17].

The long-term results of Preserflo microsurgery may be related to the early postoperative course. Transient hypotony was the most frequent postoperative complication in our study (17%). One of these patients had choroidal effusion, a shallow anterior chamber, and decreased visual acuity. The design of the Preserflo MicroShunt obeys the assumptions of the Hagen–Poiseuille equation for the prediction of IOP. Therefore, if the aqueous production exceeds 2 μL/min, postoperative IOP should be maintained above 5 mmHg. However, there are other risk factors for hypotony, such as the washout period of IOP-lowering medications. Beckers et al. presented results from a 2–4-year multicenter study and reported transient hypotony in 16.3% (7/43) of patients treated with 0.4 mg/mL MMC and 0% of patients treated with 0.2 mg/mL MMC [18]. Battle et al. reported transient hypotony in 13% of patients (3/23) and transient choroidal effusion in 8.7% (2/23) of all operated patients treated with 0.4 mg/mL MMC [10]. In our study, 3 of 4 patients with hypotony received MMC at a concentration of 0.2 mg/mL. Pillunat et al. reported statistically significantly more transient postoperative hypotony cases (69%) in the Preserflo group than in the trabeculectomy group (27%) [8]. Transient numerical hypotony is the most common complication and can be managed with observation. If shallowing of the anterior chamber or large choroidal effusion is observed, cycloplegics can be commenced, and if necessary, the injection of viscoelastic into the anterior chamber can be performed [9]. The primary cause of the choroidal effusion is hypotony, and outcomes can be good with prompt recognition and management [19].

Other risk factors for choroidal effusion include sudden IOP decrease during surgery, previous ocular surgery, high myopia, inflammation, use of blood anticoagulants, atherosclerosis, older age, and eye trauma. In the case of our patient who developed choroidal effusion, previous ocular surgery (Ex-Press Mini Glaucoma Shunt and trabeculoplasty), older age (84 years old), and atherosclerosis were noted as risk factors.

Keratitis in our study was noticed in two patients (6.7%). Beckers et al. evaluated 81 patients with POAG and reported mild-to-moderate keratitis cases (11.1%) [18]. In our study, two cases of keratitis healed within 7 days and their visual acuity improved. Both subjects had received 0.5 mg/mL MMC. We suspected that the dose of MMC and the manner of its placement (not deep enough under Tenon’s capsule and at the conjunctiva margin in contact with the limbus) were the main causes of this complication.

Studies have shown bleb failure with an IOP increase in MicroShunt patients ranging from 32% to 37.7% [16,20,21]. Beckers et al. showed a low rate of postoperative needling procedures after MicroShunt implantation as follows: 6 of 81 subjects (6.2%) [18], in contrast to the rate of needling performed after trabeculectomy (14%) in the study by Gedde et al. [21]. In our study, 13.3% of subjects experienced increased IOP, 6.7% required needling at 2 weeks after surgery, and 3.3% revision at 1 month postoperatively when needling failed.

Despite the described complications, the Preserflo MicroShunt seems to be an effective surgical device for glaucoma treatment. Wagner et al. reported no statistically significant differences between trabeculectomy, XEN45 gel stent, and Preserflo MicroShunt in terms of surgical success after 6 months, but the reduction in IOP was significantly greater in the trabeculectomy group [22]. Fea et al. showed the IOP-lowering effect of the Preserflo MicroShunt, which reduced the need for antiglaucoma medications in patients with POAG and PXG. They found no significant differences between POAG and PXG eyes, or between phakic and pseudophakic eyes, when comparing the mean IOP-lowering effect. Regarding complications, the most common adverse events were hyphema (7.7%) and choroidal detachment (4.8%). These were mild in severity and were resolved successfully, which was similar to the findings of our study. Needling was performed in 18.3% and surgical revision in 13.5% of eyes, which were higher than the rates in our study [23]. In a study by Scheres et al., microshunts required fewer bleb interventions (10%) than were required with Xen45 gel stent treatment (24%), but hypotony at any time was more frequent (39%) in the microshunt group than in the Xen45 gel stent group (24%). Interestingly, choroidal detachment occurred equally (2%) in both procedures [24]. Battle et al. [25] observed transient hypotony in 17.4% of microshunt cases, which resolved within 8 (±0.0) days, a flat anterior chamber in 8.7% of cases, which resolved within 15 (±0.0) days, bleb-related complications in 13% of cases, which resolved within 47.3 (±63.1) days, and hyphema in 8.7% of cases, which healed within 8.5 (±10.6) days.

Although the rate of early complications in our study was 46.7%. Similarly, to other MIGS devices that aim for subconjunctival filtration, bleb-related problems are the first to need to be resolved. Bleb fibrosis and encapsulation remain the most common causes of surgical failure, despite the use of MMC. It requires post-operative needle revision, which in most cases enables the restoration of bleb functioning. Other complications included transient hypotony, anterior chamber shallowing, and choroidal effusion that resolved spontaneously without additional interventions and a decrease in BCVA. Thus, the Preserflo MicroShunt, a minimally invasive glaucoma surgery device, has a good efficacy and safety profile as compared to other MIGS [23,24,25,26]. In contrast to the study of Bunod et al. [27], we did not observe any cases of tube exposure in our research.

However, this study had some limitations. First, it involved a small number of enrolled patients and lacked a control group. The results showed significant comparison after 1 day of treatment, while there was no impact in other treatment groups. It may be caused by the small sample size and low statistical outcome. Moreover, there was no wash-out period, so unmedicated postoperative IOP values were compared with medicated preoperative IOP values. Therefore, in our study, we highlighted the number of complications and showed a strategy for managing them. Despite these limitations, we believe that this study is relevant in providing evidence of the safety of Preserflo MicroShunt in Caucasian populations. Further studies, particularly randomized controlled trials, comparing this device with different antiglaucoma surgeries, are needed with a longer follow-up.

## 5. Conclusions

In summary, surgical treatment plays a significant role in the management of glaucoma. This study demonstrated the early complications and management after MIGS with the Preserflo MicroShunt. It was found that Preserflo MicroShunt implantation is safe and effective for lowering IOP in patients with open-angle glaucoma. However, it is not free from transient complications. A careful patient history study seems to be the first and most important step before planning glaucoma surgery. MMC dose and postoperative control visits should be considered individually. We plan to increase the number of subjects and include corneal endothelial cell measurements in a report of the long-term results of the Preserflo MicroShunt implantation in the near future.

## Figures and Tables

**Figure 1 ijerph-19-08679-f001:**
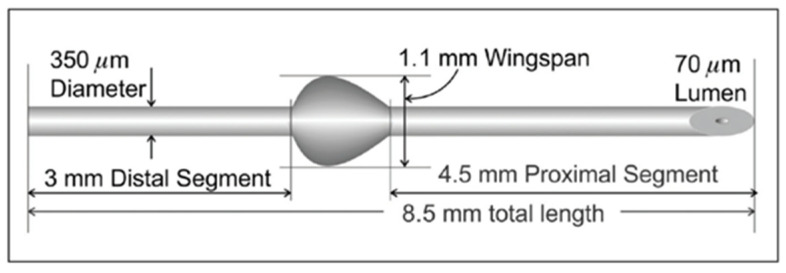
Preserflo MicroShunt (copyright Santen Pharmaceutical Co., Ltd., Osaka, Japan; reproduced with permission).

**Figure 2 ijerph-19-08679-f002:**
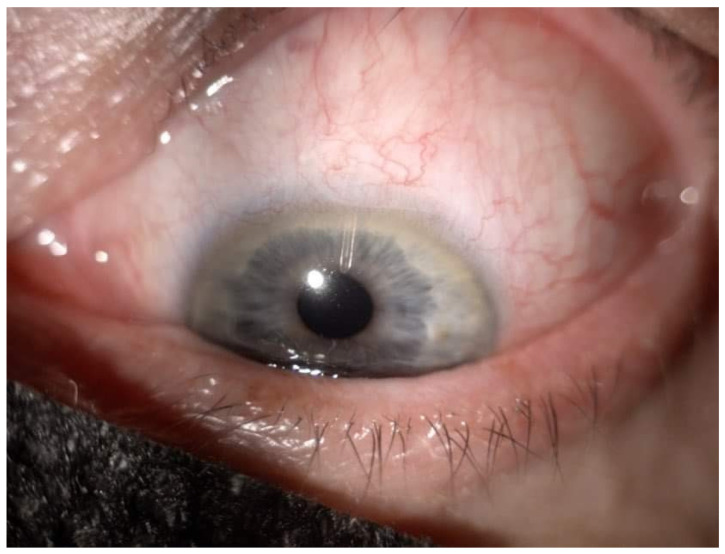
Preserflo Microshunt in anterior chamber.

**Figure 3 ijerph-19-08679-f003:**
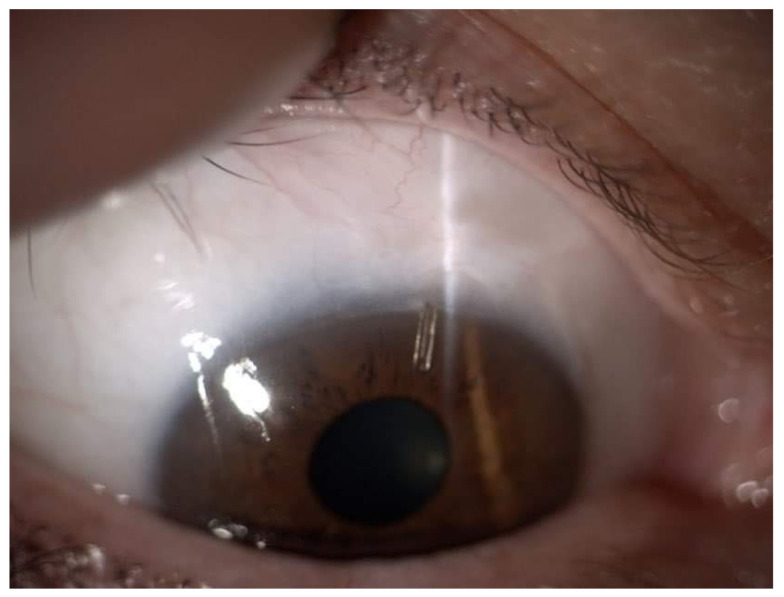
Avascular filtering bleb after Preserflo Microshunt implantation.

**Table 1 ijerph-19-08679-t001:** Group characteristic.

Characteristic	Value
Sex (female:male), *n* (%)	22 (73.3%):8 (26.7%)
Age, mean ± SD	68.53 ± 10.24
Combined treatment, *n* (%)	4 (13.3%)
Glaucoma, *n* (%)	
POAG	25 (83.3%)
PXG	5 (16.7%)
Cataract, *n* (%)	19 (63.3%)
Medications before treatment, median (Q1; Q3)	2.00 (1.25; 3.00)
MMC, *n* (%)	
0.2	22 (73.3%)
0.5	8 (26.7%)
Postoperative complications, *n* (%)	14 (46.7%)
Previous treatments, *n* (%)	
Trabeculectomy	3 (10.0%)
SLT	2 (6.7%)
Ex-Press	4 (13.3%)
Cataract surgery 3 months after Preserflo implantation	1 (3.3%)

SD, standard deviation; POAG, primary open-angle glaucoma; PEX, pseudoexfoliation glaucoma; MMC, mitomycin C; Q, quartile, SLT: selective laser trabeculoplasty.

**Table 2 ijerph-19-08679-t002:** Comparison between BCVA level before treatment and BCVA level 1 day, 2 weeks, 1 month, 3 months, and 6 months after treatment (in the whole group, in the group of people with MMC = 0.5 and MMC = 0.2).

Measurement Time	VA Level Mean ± SD or Median (Q1; Q3)	VA Level before Treatment Mean ± SD or Median (Q1; Q3)	MD (95% CI)	W/t	*p*
Whole group					
1 day after treatment	0.25 (0.10; 0.48)	0.20 (0.00; 0.040)	0.05 (<0.01; 0.30)	39.00	0.045
2 weeks after treatment	0.20 (0.10; 0.38)		0.00 (−0.10; 0.10)	79.50	0.810
1 month after treatment	0.20 (0.10; 0.30)		0.00 (−0.10; 0.05)	112.00	0.504
3 months after treatment	0.10 (0.00; 0.20)		−0.10 (−0.25; <0.01)	87.00	0.129
6 months after treatment	0.10 (0.00; 0.20)		−0.10 (−0.25; <0.01)	105.50	0.054
MMC = 0.2					
1 day after treatment	0.35 (0.10; 0.65)	0.25 (0.03; 0.40)	0.10 (−0.05; 0.40)	91.00	0.082
2 weeks after treatment	0.20 (0.20; 0.38)		−0.05 (−0.15; 0.15)	44.50	0.972
1 month after treatment	0.20 (0.10; 0.28)		−0.05 (−0.25; 0.05)	44.00	0.613
3 months after treatment	0.15 (0.00; 0.20)		−0.10 (−0.35; 0.10)	21.00	0.303
6 months after treatment	0.10 (0.00; 0.20)		−0.15 (−0.35; 0.10)	15.00	0.118
MMC = 0.5					
1 day after treatment	0.21 ± 0.17	0.15 (0.00; 0.25) 0.18 ± 0.19	0.03 (−0.05; 0.13)	1.00	0.351
2 weeks after treatment	0.20 ± 0.15		0.02 (−0.06; 0.11)	0.68	0.517
1 month after treatment	0.16 ± 0.15		−0.02 (−0.08; 0.06)	−0.42	0.685
3 months after treatment	0.14 ± 0.15		−0.04 (−0.11; 0.04)	−1.16	0.285
6 months after treatment	0.00 (0.00; 0.33)		−0.15 (−0.15; 0.10)	4.00	0.410

MD—mean/median differences (VA level after treatment minus VA level before treatment) with 95% confidence intervals. W—Wilcoxon’s test’s statistic; t—t-Student’s test’s statistic.

**Table 3 ijerph-19-08679-t003:** Comparison between IOP level before treatment and IOP level 1 day, 2 weeks, 1 month, 3 months, and 6 months after treatment (in the whole group, in the group of people with MMC = 0.5 mg/mL and MMC = 0.2 mg/mL).

Measurement Time	IOP Level Median (Q1; Q3)	IOP Level before Treatment Median (Q1; Q3)	MD (95% CI)	W	*p*
Whole group					
1 day after treatment	8.50 (6.00; 12.00)	22.50 (21.00; 26.75)	−14.00 (−17.00; −10.50)	1.50	<0.001
2 weeks after treatment	10.50 (8.25; 13.75)		−12.00 (−16.50; −10.00)	2.00	<0.001
1 month after treatment	13.00 (10.25; 18.75)		−9.50 (−13.00; −7.50)	14.00	<0.001
3 months after treatment	13.50 (12.00; 17.75)		−9.00 (−12.50; −6.50)	11.00	<0.001
6 months after treatment	14.00 (13.00; 19.00)		−8.50 (−11.50; −6.00)	16.50	<0.001
MMC = 0.2					
1 day after treatment	9.00 (6.00; 12.75)	22.50 (20.00; 26.75)	−13.50 (−18.00; −9.00)	1.50	<0.001
2 weeks after treatment	10.00 (8.25; 14.00)		−12.50 (−18.50; −9.00)	2.00	<0.001
1 month after treatment	15.50 (10.00; 21.25)		−7.00 (−13.50; −5.00)	14.00	0.001
3 months after treatment	15.50 (12.25; 19.75)		−7.00 (−13.00; −5.00)	11.00	<0.001
6 months after treatment	14.50 (13.25; 19.00)		−8.00 (−12.50; −4.50)	16.00	<0.001
MMC = 0.5					
1 day after treatment	8.00 (6.00; 10.50)	22.50 (21.75; 25.00)	−14.50 (−19.00; −11.00)	<0.01	0.014
2 weeks after treatment	12.00 (9.50; 12.00)		−10.50 (−18.00; −9.50)	<0.01	0.014
1 month after treatment	12.50 (11.75; 13.00)		−10.00 (−15.00; −8.50)	<0.01	0.014
3 months after treatment	12.00 (11.75; 13.00)		−10.50 (−14.00; −9.00)	<0.01	0.014
6 months after treatment	13.00 (12.50; 16.00)		−9.50 (−13.00; −7.00)	<0.01	0.022

MD—mean/median differences (IOP level after treatment minus IOP level before treatment) with 95% confidence intervals. W—Wilcoxon’s test’s statistic.

**Table 4 ijerph-19-08679-t004:** Range of variables.

Characteristic	Range
Age	48.00; 88.00
VA before treatment	0.00; 1.30
VA 1 day after treatment	0.00; 1.50
VA 2 weeks after treatment	0.00; 1.30
VA 1 month after treatment	0.00; 1.30
VA 3 months after treatment	0.00; 1.30
VA 6 months after treatment	0.00; 1.30
IOP before treatment	15.00; 47.00
IOP 1 day after treatment	3.00; 26.00
IOP 2 weeks after treatment	3.00; 24.00
IOP 1 month after treatment	5.00; 28.00
IOP 3 months after treatment	7.00; 26.00
IOP 6 months after treatment	8.00; 28.00

**Table 5 ijerph-19-08679-t005:** Postoperative complications/interventions.

Postoperative Complications/Interventions	*n* (%)
Hypotony	5 (16.7%)
Choroidal effusion	1 (3.3%)
Keratitis	2 (6.7%)
Flare in anterior chamber	1 (3.3%)
Increased intraocular pressure	4 (13.3%)
Needling	2 (6.7%)
Hyphema	1 (3.3%)
Preserflo Revision	1 (3.3%)

## Data Availability

All materials and information are available upon e-mail request to the corresponding author. The names and exact data of the study participants may not be available because of privacy policies.

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
