# Peer review of "Early Complications and Results of Preserflo MicroShunt in the Management of Uncontrolled Open-Angle Glaucoma: A Case Series"

_ijerph, 2022, doi:10.3390/ijerph19148679_

Round 1

Reviewer 1 Report

The present article describes the outcomes and early complications of Preserflo MicroShunt in the management of uncontrolled Open-Angle Glaucoma by monitoring 30 patients. The author made the conclusion that Preserflo MicroShunt is safe and effective for lowering the IOP but have some transient complications. Meanwhile author also mentioned about the limitations of this study which are low patient number, lack of control group and others as well that gives the initiating point for further studies in the field. The study is interesting which adds significant knowledge in the field and great efforts have been lavished by authors. However, some  minor errors that are needed to be improved are as:

1.     In abstract, either add the heading “Background” as well or remove the heading “Methods/Result/Conclusion” to make it uniform as structured style abstract with all heading or without heading, you can refer previously published paper in IJERPH.  

2.     Rewrite p-value in italics font (eg. p<0.001 as p<0.001) in whole manuscript

3.     In line 24 “Five subjects [16,7%] required anti-glau…” correct the value by removing commas as [16.7%].

4.     In line 167, correct the representation of “(p <= 0.001 for all analyses)” as “(p ≤ 0.001 for all analyses)”

5.     In line 373, the reference 3 and 4 are overlapped; with different font and style of reference 4, need to correct. 

Author Response

Responses to the Editorial Board Members' comments

Date: 14/JUL/2022

Dear Editorial Board Members and Reviewers:

We would like to thank you for the detailed review of our manuscript and your valuable remarks. The manuscript has been rechecked and the necessary changes have been made in accordance with the Reviewers’ suggestions. All changes were highlighted in red font. The responses to all comments have been prepared are given below. We hope that you will find our explanations and manuscript modifications satisfactory for reconsidering its publication in International Journal of Environmental Research and Public Health. 

Reviewer 1

The present article describes the outcomes and early complications of Preserflo MicroShunt in the management of uncontrolled Open-Angle Glaucoma by monitoring 30 patients. The author made the conclusion that Preserflo MicroShunt is safe and effective for lowering the IOP but have some transient complications. Meanwhile author also mentioned about the limitations of this study which are low patient number, lack of control group and others as well that gives the initiating point for further studies in the field. The study is interesting which adds significant knowledge in the field and great efforts have been lavished by authors. However, some  minor errors that are needed to be improved.

Response

We are grateful to the Reviewer for his insightful comments on our
manuscript. We have incorporated changes to reflect all of the suggestions provided by the Reviewer.

Comment 1

In abstract, either add the heading “Background” as well or remove the heading “Methods/Result/Conclusion” to make it uniform as structured style abstract with all heading or without heading, you can refer previously published paper in IJERPH.  

Response

We are very sorry for being inconsistent. We have removed all headings from the abstract.

Comment 2

Rewrite p-value in italics font (eg. p<0.001 as p<0.001) in whole manuscript

Response

Thank you for that constructive comment. The correction was made accordingly.

Comment 3

   In line 24 “Five subjects [16,7%] required anti-glau…” correct the value by removing commas as [16.7%].

Response

Thank you for that comment. The correction was made accordingly.

Comment 4

In line 167, correct the representation of “(p <= 0.001 for all analyses)” as “(p ≤ 0.001 for all analyses)”

Response

Thank you for that comment. The correction was made accordingly.

Comment 5

In line 373, the reference 3 and 4 are overlapped; with different font and style of reference 4, need to correct. 

Response

Thank you for that constructive comment. The correction was made accordingly.

Reviewer 2 Report

The authors describe the results of the intervention of Preserflo MicroShunt in the management of Open Glaucoma in comparison to the Gold standard techniques and highlight the possible early complications of this procedure. There are huge data available on this topic and this study further compared the characteristics of the Preserflo MicroShunt. Although, a small number of eyes are focused on this report the authors presented and discussed it nicely. There are minor suggestions and recommendations for the authors to fulfil before the final publication.

The reviewer’s comments are attached.

Author Response

Reviewer 2

The authors describe the results of the intervention of Preserflo MicroShunt in the management of Open Glaucoma in comparison to the Gold standard techniques and highlight the possible early complications of this procedure. There are huge data available on this topic and this study further compared the characteristics of the Preserflo MicroShunt. Although, a small number of eyes are focused on this report the authors presented and discussed it nicely. There are minor suggestions and recommendations for the authors to fulfil before the final publication.

Response

We are grateful to the Reviewer for his insightful comments on our
manuscript. We have incorporated changes to reflect all of the suggestions provided by the Reviewer.

Comment 1

  1.                         PreserFlo® MicroShunt (PM) (also

known as InnFocus® MicroShunt) at the first time.

Response

Thank you for that comment. The correction was made accordingly (line 52).

Comment 2

The paragraph starts from line 53, briefly gave the information of all forms of MIGS) such as                                                                                                                                                     Clemente,  CA,USA),  and  subconjunctival  stents  and  also   compare   the   features  with Preserflo MicroShunt.

Response

Thank you for that comment. Following description has been added (lines 51-55).

“Among MIGS there are many devices witch different mechanism of action: Schlemm’s Canal implants that correct conventional outflow such as iStent®, Hydrus® ; suprachoroidal space implants that strengthen outflow via the unconventional route: CyPass®, iStent Supra® ; subconjunctival implants that bypass physiological aqueous drainage pathways: XEN® Gel Stent, Preserlo® MicroShunt”.

Comment 3

Line 80-81, mention the year of Helsinki guidelines (2008, modified 2013)

Response

Thank you for that comment. The amendment has been added (lines 84).

Comment 4

Line 86-87 write the specific criteria for selecting 30 eyes only. There are many studies whose focused hundreds of eyes.

Response

Thank you for pointing this out. We are aware that there are many studies with the larger group of patients. In our department we have just started performing the surgeries with Preserflo and we have presented all our cases (30) because we just introduce this technique to our armamentarium. The study is still pending and we plan to expand these group in the future.

Comment 5

Overall, the results showed significant comparison after 1 day of treatment, while no impact in other treatment groups. Sample size is small, this may be a cause of this low statistical outcome. Authors should discuss it to highlight the possible limitations.

Response

Thank you for that comment. Following description has been added (lines 342-344):

Overall, the results showed significant comparison after 1 day of treatment, while no impact in other treatment groups. Sample size is small, this may be a cause of this low statistical outcome.

Comment 6

In table 1, with Age, mean ± SD (years) should be represented, give the abbreviation of SLT in the bottom of table 1. There is no demographic data presented for male eyes in table 1.

Response

We are sorry for misleading. In table 1 the age has been represented as follows: 68.53 ± 10.24. The abbreviation of SLT was added as follows: selective laser trabeculoplasty. The demographic data was presented for whole cohort without dividing for gender, the clarification has been added.

Comment 7

Response

We are sorry for overlooking. The units have been added.

Comment 8

Authors should provide a sample image with a Slit-lamp photograph to present the PreserFlo MicroShunt effect on the eye.

Response

Thank you for that suggestion, the figures 2 and 3 have been added.

Comment 9

In discussion, the improvement is suggested in lieu of the review study;

Gambini G, Carlà MM, Giannuzzi F, Caporossi T, De Vico U, Savastano A, Baldascino A, Rizzo C, Kilian R, Caporossi A, Rizzo S. PreserFlo® MicroShunt: An Overview of This Minimally Invasive Device for Open-Angle Glaucoma. Vision (Basel). 2022 Feb 9;6(1):12. doi: 10.3390/vision6010012

Response

Thank you for bringing this latest paper regarding Prseerflo. We have added it to our references, as well as we incorporate in discussion the following:

Similarly to other MIGS devices, that aim subconjunctival filtration, bleb-related problems are the first to need to be resolved. Bleb fibrosis and encapsulation remain the most common causes of surgical failure, despite the use of MMC. It requires post-operative needle revision, which in most cases enable to restore bleb functioning. Other complications, included transient hypotony, anterior chamber shallowing and choroidal effusion resolved spontaneously without additional interventions and without a decrease in BCVA. (lines 339-345)

Comment 10

Discuss and cite this article Bunod R, Robin M, Buffault J, Keilani C, Labbé A, Baudouin

  1. PreserFlo MicroShunt® exposure: a case series. BMC Ophthalmol. 2021 Jul 10;21(1):273. doi: 10.1186/s12886-021-02032-z. PMID: 34246229; PMCID: PMC8272321.

Response

Thank you for that comment. Following description has been added (lines 347-348).

Comment 11

The conclusion should be revised and should be focused on the outcome of the present study. There is no need to state the background in this section.

Response

The conclusions were rewritten according to the Reviewer suggestion. (lines 360-368)

This manuscript is a resubmission of an earlier submission. The following is a list of the peer review reports and author responses from that submission.

Round 1

Reviewer 1 Report

In the reviewed article the authors provide an insight in the postoperative follow-up of 30 patients operated with Preserflo MicroShunt. An analysis of BCVA and IOP is provided, which is grouped by different MMC dosages . A detailed description of postoperative complications is given. 

While it is interesting, that the authors share their experiences with this new device, the aim and purpose of the article are not clearly indicated. As the authors have stated there is no control group. Thus, the conclusions are not adequately supported by the findings. In- and exclusion criteria have to be defined. It is difficult to interpret BCVA results when some patients had combined operations with phacoemulsification. The findings of this mansucript could rather be presented as a case series.

Line 90

Thirty Caucasian patients (22 women and 8 men) in the mean age ....

This sentence should be in the results section.

Line 109

withdrawn instead of withdraw

Line 240 

insert "IOP"

Discussion section

Reference numbers in the discussion section have to be revised.